# Motion Graph Unleashed:
# A Novel Approach to Video Prediction

**Yiqi Zhong**[13*]    **Luming Liang**[1*]    **Bohan Tang**[2]    **Ilya Zharkov**[1]    **Ulrich Neumann**[3]

[1]Microsoft    [2]University of Oxford    [3]University of Southern California

[1]{yiqizhong,lulian,zharkov}@microsoft.com

[2]bohan.tang@eng.ox.ac.uk    [3]{yiqizhon,uneumann}@usc.edu

## Abstract

We introduce *motion graph*, a novel approach to the video prediction problem, which predicts future video frames from limited past data. The motion graph transforms patches of video frames into interconnected graph nodes, to comprehensively describe the spatial-temporal relationships among them. This representation overcomes the limitations of existing motion representations such as image differences, optical flow, and motion matrix that either fall short in capturing complex motion patterns or suffer from excessive memory consumption. We further present a video prediction pipeline empowered by motion graph, exhibiting substantial performance improvements and cost reductions. Experiments on various datasets, including UCF Sports, KITTI and Cityscapes, highlight the strong representative ability of motion graph. Especially on UCF Sports, our method matches and outperforms the SOTA methods with a significant reduction in model size by $78\%$ and a substantial decrease in GPU memory utilization by $47\%$. Please refer to this link for the official code.

## 1 Introduction

Video prediction aims at predicting future frames given a limited number of past frames. This technology has potential for numerous applications, including video compression, visual robotics, and surveillance systems. A critical aspect of designing an effective video prediction system is the precise modeling of motion information, which is essential for its successful deployment.

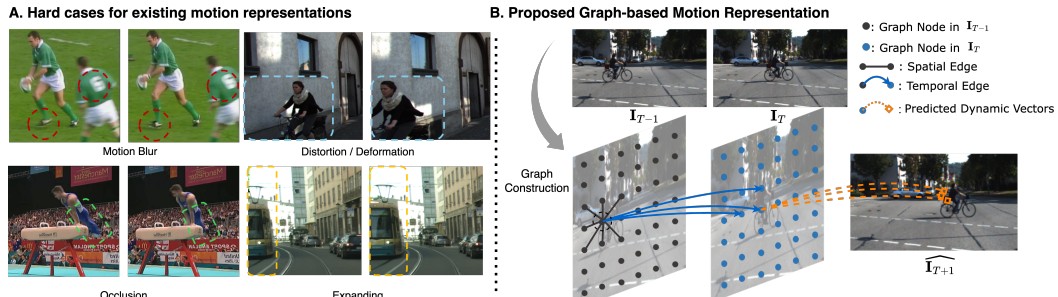

Figure 1: (A) Hard cases which cannot be properly modeled by most existing representations. (B) Motion graph transforms single-frame patches into interconnected nodes, describing the spatial-temporal relationships. Future per-pixel motion dynamic vectors are then predicted on this graph.

---

[1]Equal contributions. The work is mostly done during Yiqi Zhong's internship at Microsoft.

38th Conference on Neural Information Processing Systems (NeurIPS 2024).

At first glance, video prediction appears as a sequence prediction problem, with conventional approaches using advanced sequential data modeling techniques such as 3D convolution [1, 2], recurrent neural networks [3, 4, 5, 6, 7], and transformers [8, 9, 10]. These methods implicitly model motion and image appearance to help reason sequence evolution. However, the implicit approach to motion modeling can result in inefficient learning, which stems from the inherent difference between video prediction and typical sequence prediction: in typical sequence prediction, features tend to evolve uniformly; whereas video prediction involves not only static elements like object appearance but also dynamic elements intricately embedded in the sequence, e.g., motion-related information, including object pose and camera movement. Effectively managing both the static and dynamic aspects of the hidden feature space presents a great challenge.

Instead of treating video prediction purely as a sequence prediction problem, an alternative approach is to explicitly model inter-frame motion, and redefine the task primarily as a motion prediction problem [11, 12, 13, 10, 2]. For this approach, it is crucial to select an appropriate motion representation, which should exhibit expressiveness by accurately encapsulating motion with detailed information like direction and speed. However, as Section 2 and Table 1 will elaborate, existing video motion representations have limitations. Some, as Figure 1(A) demonstrates, lack representative ability for complex scenarios. For example, optical flow is unable to handle motion blur and object distortion/deformation resulting from perspective projection. Others over-sacrifice memory efficiency to model complex motion patterns, such as the motion matrices proposed in MMVP [2].

To fill the gap in existing research, we propose a novel motion representation: the motion graph, which treats image patches of the observed video frames in the downsample space as interconnected graph nodes based on their spatial and temporal proximity; See Figure 1 (B) for an illustration. This approach achieves both goals of representativeness and compactness by generating accurate predictions while conserving memory and computational resources. For the representative ability, our motion graph enriches each node with dynamic information, including multiple feasible weighted flows toward the subsequent frame. This method captures a broader range of motion patterns compared to single-direction representations like optical flow and enhances error tolerance. Additionally, the motion graph excels in compactness by avoiding computation-heavy operations such as stacked convolutional layers. To reduce computational complexity, we also limit the connections in the motion graph to a fixed number of neighbors for each node. Consequently, compared to previous representations (e.g., motion matrix, keypoint heatmap), the motion graph offers a more sparse and memory-efficient modeling of motion, effectively transcending the limitations of 2D image structures.

Based on the motion graph we have introduced, we present a novel video prediction pipeline. The representative ability and compactness of the motion graph allow our system to have achieved notable enhancements in computational efficiency. We test various scenarios with different motion patterns on three well-known datasets, UCF Sports, KITTI, and Cityscapes. Across all three datasets, our approach consistently exhibits performance that aligns with or surpasses the state-of-the-art (SOTA) methods with significantly lower GPU comsumption.

## 2   Related Works

In this section, we mainly discuss the existing video prediction systems that apply to the setting of short-term high resolution video prediction, please see more details about the setting in Section A.5.

Based on whether they explicitly model motion, existing video prediction methods can be put into two categories. Methods that do not explicitly model motion usually consider video prediction purely as a sequence prediction problem [3]. They use advanced sequential modeling techniques such as recurrent neural networks [4, 3, 5, 14, 15, 16, 17, 18] and transformers [8, 10, 19, 20] to estimate the evolution of image features in the hidden space along the temporal dimension. However, video prediction holds a unique property that is unlike text generation or other sequence prediction problems: that is, most features in video sequences (e.g., object appearance, scene layout, texture of the background) usually remain unchanged, or have no major swift change, from frame to frame. Yet, sequential modeling techniques treat the whole image features uniformly; using them for video prediction thus requires modification in the inner structure of certain sequential modeling architectures to accommodate this property of video prediction. Some works choose to enhance the spatial consistency by adding complex residual shortcuts [4, 17, 21], or by decomposing temporal and spatial information into two separate hidden spaces, hoping that the network will figure out which to maintain and which

to change [15, 16]. Although such accommodations sometimes help systems reach impressive prediction accuracy (given the powerful sequential modeling capability of the advanced techniques), these methods tend to have larger model sizes and complicated architectures.

To improve the efficiency of motion modeling in systems, this paper extends the idea shared by the second type of methods, which treat video prediction as a motion prediction problem, and explicitly model the motion hidden in image sequences. Such methods either reason future motion patterns using certain representations with image features as the input, or solely use motion representations as the input for prediction. There are four lines of work on motion representations. First, image difference [11, 10] uses the subtraction of two consecutive frames to represent the dynamic of image sequences. It requires almost no computational cost to calculate but does not directly describe motion. Second, optical flow/ voxel flow [22, 23, 24, 13] describes the pixel-level motion. But, it usually requires auxiliary or out-of-shell models to generate, and is limited to only modeling one-to-one relationships [2], which can be a disadvantage, especially for scenarios with ambiguous pixel-to-pixel correspondence (e.g., motion blur). Third, key point trace [12] serves highly structured scenarios (e.g., videos with static backgrounds and dominated by human motions) but can be inefficient for more complex content, including crowded scenes with multiple objects. Fourth, the recent motion matrix [2] describes the all-pair relationship between consecutive frames to overcome the drawback of optical flow. While motion matrix supports a more efficient and compact system of a notably smaller size, concerns were over the computational and space complexity of its operations [2].

Table 1: Motion representation comparison. We assess the representative ability by judging how accurate the motion can be described, especially for the hard cases demonstrated in Fig 1(A).

| Motion Representation | Out-of-shell Model | Representative Ability | Space Complexity |
|---|---|---|---|
| Image difference [11, 10] | n/a | low | $O(n)$ |
| Keypoint trace [12] | required | medium | $O(n)$ |
| Optical flow/ voxel flow [22, 23, 24, 13] | Some required | medium | $O(n)$ |
| Motion matrix [2] | n/a | high | $O(n^2)$ |
| Motion graph (*ours*) | n/a | high | $O(n)$ |

Besides the above motion representations, we notice that graph, as a sparse data structure that can accurately describe the relationships between entities, has been widely adopted in ordinary motion prediction systems, such as trajectory forecasting [25, 26, 27] and human motion prediction [28, 29, 30]. Inspired by these successful adoptions, we propose to represent the motion in videos using graph structures, named *motion graph* in this work.

The major difference between motion graph and previous motion representations is that motion graph can describe many-to-many temporal and spatial correspondence with low space complexity. Such sparse representations not only enhance representative ability but also advantageously i) replace most 2D convolution operations by graph operations which are mostly implemented by linear layers, leading to faster and more parameter-efficient systems; ii) enable convenient spatial and temporal interactions among feature patches, promoting higher learning efficiency and prediction accuracy.

## 3 Methodology

In this work, video prediction is approached as a motion prediction problem, utilizing a novel representation called the motion graph. This representation is designed to effectively capture the intricate motion dynamics within video sequences. Leveraging the motion graph, we develop a video prediction pipeline that achieves high performance with reduced model size and GPU memory requirements. Section 3.1 details our problem formulation and notation. Section 3.2 explains the construction of the motion graph from input video sequences. Finally, Section 3.3 delves into our video prediction pipeline, illustrating how the motion graph is used for effective video prediction.

### 3.1 Problem Formulation

Given a video sequence with $T$ frames $\{\mathbf{I}_t \in \mathbb{R}^{H \times W \times 3} | t = 0, 1, ..., T - 1\}$, a video prediction system aims to predict the next $T'$ frames $\{\mathbf{I}_{t'} \in \mathbb{R}^{H \times W \times 3} | t' = T, T + 1, ..., T + T' - 1\}$. We approach this task by regarding video prediction as a pixel-level motion prediction problem.

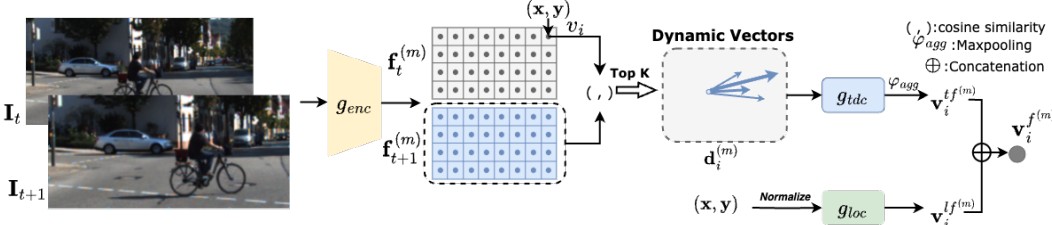

Figure 2: **Motion graph node construction**: Cosine similarity, denoted by $(,)$, between patch features in consecutive frames is computed to further choose top $k$ directions for each patch. Tendency $\mathbf{v}_i^{tf^{(m)}}$ and location features $\mathbf{v}_i^{lf^{(m)}}$ are then generated based on these $k$ vectors and the patch location.

Consider a pixel located at $(x, y)$ in a given video frame $\mathbf{I}_t$. Our system is designed to predict $k$ dynamic vectors starting from this pixel, pointing to the possible locations in the target future frame $\mathbf{I}_{t'}$. These vectors represent the pixel's anticipated motion from its current position in $\mathbf{I}_t$ to its future position in $\mathbf{I}_{t'}$. Mathematically, we express the dynamic vectors of all pixels across the observed frames as $\mathbf{P} \in \mathbb{R}^{T \times H \times W \times k \times 3}$, where $\mathbf{P}_{t,x,y} = \left[ [\Delta x_1, \Delta y_1, w_1], ..., [\Delta x_k, \Delta y_k, w_k] \right]$ with both the motion direction $(\Delta x_k, \Delta y_k)$ and the weighted component $(w_k)$. The transformation of previous frames into future frames is performed by a pixel-level image warper, denoted as $\overrightarrow{\mathcal{W}}$. The warping is based on the predicted dynamic vectors and is formally defined by the following equation:

$$\widehat{\mathbf{I}_T} = \overrightarrow{\mathcal{W}}(\mathbf{P}, \mathbf{I}_0, ..., \mathbf{I}_{T-1}). \tag{1}$$

This formulation allows us to capture and translate the intricate motion dynamics within the video sequence into future frame predictions with enhanced accuracy.

### 3.2 Motion Graph

#### 3.2.1 Intuition

Inspired by advancements in general motion prediction systems [25, 26, 30], we posit that modeling semantic correlations among image patches within observed frames is essential for improving current motion representations in video prediction. To capture such correlations, we first model the semantic information of each image patch across a video with $T$ observed frames. This is achieved by multi-scale image patch feature mappings, $\mathcal{F} = \{\mathbf{f}^{(1)}, \cdots, \mathbf{f}^{(M)}\}$, generated by inputting each observed frame into a shared image encoder $g_{enc}$ and applying a pixel unshuffle technique [31] to reshape all $M$ feature maps to the resolution of the smallest features. Here, the smallest resolution of the features generated by $g_{enc}$ is $H_s \times W_s$, and $\mathbf{f}^{(m)} \in \mathbb{R}^{T \times H_s \times W_s \times C_m}$ represents the feature map in the $m$-th scale containing the $C_m$-dimensional features of $T \times H_s \times W_s$ distinct image patches.

Based on $\mathcal{F}$, we construct a multi-view graph, named *motion graph*, to capture the semantic correlations among $T \times H_s \times W_s$ image patches. This graph is represented as: $\mathcal{G} = \{\mathcal{V}, \mathcal{E}^{(1)}, \cdots, \mathcal{E}^{(M)}\}$, where $\mathcal{V} = \{v_0, \cdots, v_{TH_sW_s-1}\}$ denotes the set of nodes each corresponding to an image patch, and $\mathcal{E}^{(m)}$ denotes the edge set for the $m$-th view capturing the correlations driven by the semantic information within $\mathbf{f}^{(m)}$. We detail the graph construction and the graph-related operations as follows.

#### 3.2.2 Node Motion Feature Initialization

For each node $v_i$ in the $m$-th view, we initialize its motion feature $\mathbf{v}_i^{f^{(m)}} \in \mathbb{R}^{d_{mf}^{(m)}}$ with two components: the tendency feature $\mathbf{v}_i^{tf^{(m)}} \in \mathbb{R}^{d_{tf}^{(m)}}$ and the location feature $\mathbf{v}_i^{lf^{(m)}} \in \mathbb{R}^{d_{lf}^{(m)}}$. The tendency feature $\mathbf{v}_i^{tf^{(m)}}$ captures the node's motion-related attributes relative to nodes in the subsequent frame and the location feature $\mathbf{v}_i^{lf^{(m)}}$ models the normalized absolute location of each node in a frame.

Inspired by prior works [2, 32], we generate the tendency feature $\mathbf{v}_i^{tf^{(m)}}$ in three steps. First, for each node $v_i$, we form a node list $\mathcal{L}_i^{(m)}$ by choosing top-$K$ scoring image patches from the subsequent frame based on the cosine similarities computed by $\mathbf{f}_{\lfloor \frac{i}{H_sW_s} \rfloor}^{(m)}$ and $\mathbf{f}_{\lfloor \frac{i}{H_sW_s} \rfloor + 1}^{(m)}$. By doing so, we mitigate

the risk of false positives and better accommodate complex motion patterns, especially in downscaled spaces where an image patch may exhibit multiple potential movements. Second, we define the dynamic vector for each node $v_i$ as $\mathbf{d}_i^{(m)} = [\Delta x_1, \Delta y_1, w_1, \cdots, \Delta x_K, \Delta y_K, w_K] \in \mathbb{R}^{3K}$, where $\Delta x_k, \Delta y_k$ indicate the motion direction and $w_k$ is the cosine similarity score between $v_i$ and the $k$-th node in $\mathcal{L}_i^{(m)}$. Here, an exception is made for nodes in the last observed frame $\mathbf{I}_{T-1}$, where we apply zero-padding to the dynamic vectors, as $\mathbf{I}_T$ is unknown. Finally, we generate the tendency feature $\mathbf{v}_i^{tf^{(m)}}$ for each node by feeding each dynamic vector $\mathbf{d}_i^{(m)}$ into a multilayer perceptron (MLP) $g_{tdc}(.)$ followed by a max-pooling operation $\varphi_{agg}(.)$, as shown in:

$$\mathbf{v}_i^{tf^{(m)}} = \varphi_{agg}(g_{tdc}(\mathbf{d}_i^{(m)})). \tag{2}$$

The location feature $\mathbf{v}_i^{lf^{(m)}}$ encodes the position of a pixel in a frame. Pixel positions influence motion patterns. For instance, pixels on the sides of a frame may appear to move differently than pixels in the center for street views collected by wide-range moving cameras, due to the perspective projection effect. We use another MLP $g_{loc}(.)$ to extract the location feature and define it as:

$$\mathbf{v}_i^{lf^{(m)}} = g_{loc}\left(\frac{x}{W_s}, \frac{y}{H_s}\right), \tag{3}$$

Finally, the motion features of each node $\mathbf{v}_i^{f^{(m)}}$ are initialized as follows:

$$\mathbf{v}_i^{f^{(m)}} = \oplus(\mathbf{v}_i^{lf^{(m)}}, \mathbf{v}_i^{tf^{(m)}}), \tag{4}$$

where $\oplus(\cdot)$ denotes the concatenation operation. Figure 2 details the node motion feature initialisation.

### 3.2.3 Edge Construction

After initializing the node features, we construct edges in the motion graph to capture the semantic relationships between image patches in the observed frames. We further represent the edge set of the $m$-th view as $\mathcal{E}^{(m)} = \{\mathcal{E}^{S^{(m)}}, \mathcal{E}^{B^{(m)}}, \mathcal{E}^{F^{(m)}}\}$, where $\mathcal{E}^{S^{(m)}}$ is spatial edges, $\mathcal{E}^{B^{(m)}}$ is backward edges, and $\mathcal{E}^{F^{(m)}}$ is forward edges. Generally, spatial edges are posit on that neighboring image patches in a frame likely influence each other's future motion, and backward and forward edges connect nodes across adjacent frames indicating potential motion paths. Notably, nodes in the first frame are not assigned backward edges, and nodes in the last frame are not assigned forward edges.

We construct these edges in two steps: 1) finding the neighbors of each node connected by spatial, backward, and forward edges respectively; and 2) generating the three types of edges by connecting the neighboring nodes found. For the first step, we denote $\mathcal{N}_i^{S^{(m)}}$, $\mathcal{N}_i^{B^{(m)}}$, and $\mathcal{N}_i^{F^{(m)}}$ as sets containing neighbors of $v_i$ connected by the spatial, backward and forward edges, respectively. Then, the neighbors of each node $v_i$ are found by solving the following optimization problems:

$$\mathcal{N}_i^{S^{(m)}} = \arg\max_{\Omega}||\mathbf{C}_{i,\Omega}^{(m)}||_{1,1} \;\; s.t. \;\; \Omega \subseteq \mathcal{S}_i, \;\; |\Omega| = k,$$

$$\mathcal{N}_i^{B^{(m)}} = \arg\max_{\Omega}||\mathbf{C}_{i,\Omega}^{(m)}||_{1,1} \;\; s.t. \;\; \Omega \subseteq \mathcal{B}_i, \;\; |\Omega| = k, \tag{5}$$

$$\mathcal{N}_i^{F^{(m)}} = \arg\max_{\Omega}||\mathbf{C}_{i,\Omega}^{(m)}||_{1,1} \;\; s.t. \;\; \Omega \subseteq \mathcal{F}_i, \;\; |\Omega| = k,$$

where $\mathbf{C}^{(m)} \in \mathbb{R}^{TH_sW_s \times TH_sW_s}$ contains cosine similarity scores between nodes in $\mathcal{V}$ computed by $\mathbf{f}^{(m)}$, $k \in \mathbb{Z}$ is a hyperparameter, $\mathcal{S}_i = \{v_j \in \mathcal{V} : \lfloor\frac{i}{H_sW_s}\rfloor = \lfloor\frac{j}{H_sW_s}\rfloor\}$, $\mathcal{B}_i = \{v_j \in \mathcal{V} : \lfloor\frac{i}{H_sW_s}\rfloor = \lfloor\frac{j}{H_sW_s}\rfloor + 1\}$, and $\mathcal{F}_i = \{v_j \in \mathcal{V} : \lfloor\frac{i}{H_sW_s}\rfloor = \lfloor\frac{j}{H_sW_s}\rfloor - 1\}$. For the second step, we construct $\mathcal{E}^{S^{(m)}}$, $\mathcal{E}^{B^{(m)}}$, and $\mathcal{E}^{F^{(m)}}$ by connecting each $v_i$ with nodes in $\mathcal{N}_i^{S^{(m)}}$, $\mathcal{N}_i^{B^{(m)}}$, and $\mathcal{N}_i^{F^{(m)}}$ respectively.

### 3.2.4 Motion Graph Interaction Module

After constructing the nodes and establishing the edges in the motion graph, we enable information flow within the $m$-th view of the graph via the message-passing operation $g_{mp}$ defined as follows:

$$\mathbf{v}'^{(m)} = g_{mp}\left(\mathbf{v}^{(m)}, \mathcal{E}^{in^{(m)}}\right), \tag{6}$$

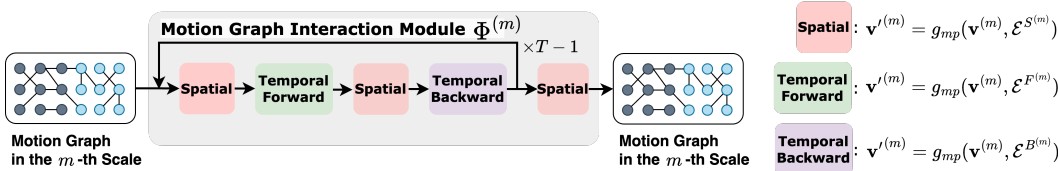

Figure 3: **Inside the interaction module for the** $m$**-th view** $\Phi^{(m)}$. The spatial and temporal message passing are iteratively conducted and repeated $T-1$ times, where $T$ is the observed frame number.

where $\mathbf{v}^{(m)} \in \mathbb{R}^{TH_sW_s \times d^{(m)}}$ is the input node features, $\mathbf{v}'^{(m)} \in \mathbb{R}^{TH_sW_s \times d^{(m)}}$ is the updated node features, and $\mathcal{E}^{in^{(m)}}$ denotes an edge set. The operation $g_{mp}$ facilitates the information exchange between nodes connected by edges in $\mathcal{E}^{in^{(m)}}$. In practice, the spatial message passing is implemented as a 2D convolution layer, while for the temporal message passing we use graph neural network to implement $g_{mp}$. See implementation details in the appendix.

With the message-passing operation $g_{mp}$, we introduce a motion graph interaction module, denoted as $\Phi$ and illustrated in Figure 3. The goal of $\Phi$ is to ensure that even nodes in the last observed frame are informed about the motion dynamics from the first observed frame, and vice versa. To achieve this comprehensive information flow, we implement $T-1$ rounds of full information transition, where $T$ is the total number of observed frames. Each round of transition includes twice the spatial message passing, once the temporal forward message passing, and once the temporal backward message passing. The combination of spatial and temporal interaction in $\Phi$ ensures holistic information integration for accurate and comprehensive motion prediction in video sequences. The design of $\Phi$ allows for thorough and balanced dissemination of motion information throughout the motion graph.

### 3.3 Motion-graph-empowered Video Prediction

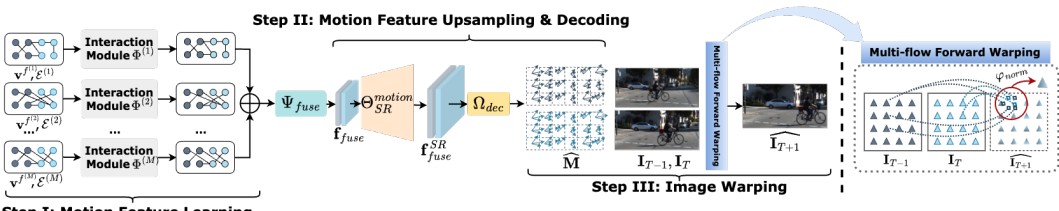

Figure 4: **Pipeline overview**. After decoding per-pixel motion features into dynamic vectors, we perform multi-flow forward warping for future frame generation.

By constructing the motion graph, we create a tool to extract motion information from the observed video frames. The following passage describes how to conduct video prediction using the constructed motion graph and its related operations. As Figure 4 indicates, to predict the unknown video frame involves three main steps, which are elaborated in the following three subsections.

#### 3.3.1 Motion Feature Learning

**Multi-view motion feature update.** After having the motion graph $\mathcal{G} = \{\mathcal{V}, \mathcal{E}^{(1)}, \cdots, \mathcal{E}^{(M)}\}$, with the node motion feature $\mathbf{v}^{f^{(m)}}$ initialized by Eq. (4) in the $m$-th view, we use the motion graph interaction module introduced in Section 3.2.4 to update $\mathbf{v}^{f^{(m)}}$ via the following formula:

$$\hat{\mathbf{v}}^{f^{(m)}} = \Phi^{(m)}\left(\mathbf{v}^{f^{(m)}}, \mathcal{E}^{(m)}\right),$$ (7)

where $\hat{\mathbf{v}}^{f^{(m)}} \in \mathbb{R}^{TH_sW_s \times d_{mf}^{(m)}}$ is the updated node features in the $m$-th view of the motion graph.

**Multi-view motion feature fusion.** Once the node feature in each view of the motion graph is updated, we concatenate the node features from each view and apply a fusion module $\Psi_{fuse}$ to integrate these multi-view features into a unified representation as follows:

$$\mathbf{f}_{fuse} = \Psi_{fuse}(\varphi_{cat}(\mathbf{v}^{f^{(m)}}|m = 1, ..., M)),$$ (8)

where $\mathbf{f}_{fuse} \in \mathbb{R}^{T \times H_s \times W_s \times C_{node}}$, with $C_{node}$ denoting the dimension of the node features.

### 3.3.2 Motion Feature Upsampling and Decoding

In this step, we transform the fused multi-view motion feature $\mathbf{f}_{fuse}$ into a 2D structure with a resolution of $H_s \times W_s$. This feature is then upscaled to match the resolution of the original video frames ($H \times W$) using a motion upsampler, $\Theta_{SR}$. The architecture of $\Theta_{SR}$ is inspired by ResNet-based image super-resolution networks [33], which progressively refine features from lower to higher resolutions until reaching the original image size. This upsampling process is key to predicting pixel-level motion features for all observed frames, which extend toward the next future frame. By incrementally adjusting the resolution, $\Theta_{SR}$ effectively bridges the gap between the multi-view motion information and the high-resolution requirements of accurate pixel-level motion prediction.

Upon obtaining the pixel-level motion feature $\mathbf{f}_{fuse}^{SR}$, we use a motion decoder, $\Omega_{dec}$, to convert it into pixel-level dynamic vectors $\mathbf{P}$. As defined in Section 3.1, $\mathbf{P}$ contains $k$ potential motion directions with associated probability scores for each pixel. This decoding output is compatible with the motion features in the motion graph, which are generated by the $k$ dynamic vectors of image patches.

### 3.3.3 Image Warping

After having the dynamic vectors through the decoding process, we use them to warp the observed frames into the future frame. Unlike traditional methods that often use optical-flow-based backward warping, our approach follows the design logic of the motion graph feature learning as well as the output format to use a multi-flow forward warping technique. This method, drawing from the image splatting concepts in prior works [34, 35], is visually detailed in Figure 4. Forward warping allows each predicted dynamic vector to directly contribute to the construction of the future frame. To aggregate the contributions of multiple vectors at each pixel location in the synthesized frame, we apply a normalization operation $\varphi_{norm}$. It normalizes the weight of each vector based on the sum of their weights, ensuring an even and balanced contribution to the final pixel value in the future frame.

## 4 Experiments

For evaluation, we trained our video prediction pipeline in an end-to-end fashion on three public datasets: i) **UCF Sports** [36] (i.e. 150 video sequences emphasizing various sports scenes, with frame resolution, and two different data splits to date—one from STRPM [15], one from MMVP [2]—evaluation is on both splits); ii) **KITTI** [37] (i.e. 28 driving videos with a resolution of $375 \times 1242$; following previous works [38, 13], the image frames are resized to $256 \times 832$); and iii) **Cityscapes** [39] (i.e. 3,475 driving videos with 2,945 in the training set and 500 in the validation set). For each dataset, by default, we set $k = max(10, 1\% \times H_s \times W_s)$ which is $1\%$ of the smallest feature map's resolution and no larger than 10 for efficiency consideration. Table 2 shows the configuration for each dastaset. More training and implementation details are in the appendix.

Table 2: Dataset configurations

| Dataset | Resolution | $H_s$ | $W_s$ | Input Frame | Output Frame | Training Loss | k |
|---|---|---|---|---|---|---|---|
| UCF Sports | $512 \times 512$ | 32 | 32 | 4 | 1 | mean-square error (following [15, 16, 2]) | 10 |
| KITTI | $256 \times 832$ | 16 | 52 | 2 | 1 | l1 + Perceptual loss(following [13]) | 8 |
| Cityscapes | $512 \times 1024$ | 32 | 64 | 2 | 1 | l1 + Perceptual loss (following [13]) | 10 |

### 4.1 Public Benchmark Comparison

On the UCF Sports STRPM split, we evaluate the method on two metrics following previous research [15]: peak signal-to-noise ratio (PSNR) and learned perceptual image patch similarity (LPIPS) [40]. The proposed method is compared with existing methods for their results of the 1st and 6th future frames ($t = 5$, $t = 10$ in Table 3). Table 3 shows that our method matches the SOTA performance on the PSNR metric and outperforms all previous methods on the LPIPS metric. On the UCF Sports MMVP split, the validation dataset has been divided into three categories: the easy (SSIM $\leq 0.9$), intermediate ($0.6 \leq$ SSIM $< 0.9$), and hard subsets (SSIM $< 0.6$), which take up $66\%$, $26\%$, and $8\%$ of the full set respectively. We evaluate the methods on PSNR, LPIPS, and Structural Similarity Index Measure (SSIM). Table 4 shows that the proposed method outperforms all other methods.

Table 3: Performance comparison on UCF Sports STRPM split

| Method | $t=5$ | | $t=10$ | |
| --- | --- | --- | --- | --- |
| | PSNR ↑ | LPIPS$_{\times100}$ ↓ | PSNR ↑ | LPIPS$_{\times100}$ ↓ |
| ConvLSTM (NeurIPS 2015) [3] | 26.43 | 32.20 | 17.80 | 58.78 |
| BeyondMSE (ICLR 2016) [41] | 26.42 | 29.01 | 18.46 | 55.28 |
| PredRNN (NeurIPS 2017) [4] | 27.17 | 28.15 | 19.65 | 55.34 |
| PredRNN++ (ICML 2018) [5] | 27.26 | 26.80 | 19.67 | 56.79 |
| SAVP (arXiv 2018) [42] | 27.35 | 25.45 | 19.90 | 49.91 |
| SV2P (ICLR 2018) [43] | 27.44 | 25.89 | 19.97 | 51.33 |
| E3D-LSTM (ICLR 2019) [17] | 27.98 | 25.13 | 20.33 | 47.76 |
| CycleGAN (CVPR 2019) [44] | 27.99 | 22.95 | 19.99 | 44.93 |
| CrevNet (ICLR 2020) [45] | 28.23 | 23.87 | 20.33 | 48.15 |
| MotionRNN (CVPR 2021) [7] | 27.67 | 24.23 | 20.01 | 49.20 |
| STRPM (CVPR 2022) [15] | 28.54 | 20.69 | 20.59 | 41.11 |
| STIP (arXiv 2022) [16] | 30.75 | 12.73 | 21.83 | 39.67 |
| DMVFN (CVPR 2023) [13] | 30.05 | 10.24 | 22.67 | 22.50 |
| MMVP (ICCV 2023) [2] | 31.68 | 7.88 | 23.25 | 22.24 |
| Ours | **31.70** | **6.61** | **23.27** | **19.94** |

Table 4: Performance comparison on the UCF Sports MMVP split.

| Method | Full set | | | Easy (SSIM ≥ 0.9) | | | Intermediate (0.6 ≤ SSIM < 0.9) | | | Hard (SSIM < 0.6) | | | Model Size↓ |
| --- | --- | --- | --- | --- | --- | --- | --- | --- | --- | --- | --- | --- | --- |
| | SSIM ↑ | PSNR ↑ | LPIPS ↓ | SSIM ↑ | PSNR ↑ | LPIPS ↓ | SSIM ↑ | PSNR ↑ | LPIPS ↓ | SSIM ↑ | PSNR ↑ | LPIPS ↓ | |
| STIP [16] | 0.8817 | 28.17 | 0.1626 | 0.9491 | 30.65 | 0.1066 | 0.8351 | 23.97 | 0.2271 | 0.4673 | 15.97 | 0.4450 | 18.05M |
| SimVP [1] | 0.9189 | 29.97 | 0.1326 | 0.9664 | 32.87 | 0.0584 | 0.8845 | 25.79 | 0.1951 | 0.6267 | 18.99 | 0.5600 | 3.47M |
| MMVP [2] | 0.9300 | 30.35 | 0.1062 | 0.9674 | 33.05 | 0.0580 | 0.8970 | 26.29 | 0.1569 | 0.7203 | **20.84** | 0.3510 | 2.79M |
| Ours | **0.9314** | **30.49** | **0.0823** | **0.9685** | **33.23** | **0.0444** | **0.8978** | **26.36** | **0.1348** | **0.7264** | 20.83 | **0.2320** | **0.60M** |

Table 5: Evaluation on Cityscapes [39] and KITTI [37] datasets. "RGB", "F", "S" and "I" denote video frames, optical flow, semantic map, and instance map; $t+3$ and $t+5$ respectively indicate the average performance of the next 3 and 5 frames. Results with * are copied from [13].

| Method | Input | Cityscapes | | | | | | KITTI | | | | | |
| --- | --- | --- | --- | --- | --- | --- | --- | --- | --- | --- | --- | --- | --- |
| | | MS_SSIM($\times10^{-2}$)↑ | | | LPIPS($\times10^{-2}$)↓ | | | MS_SSIM($\times10^{-2}$)↑ | | | LPIPS($\times10^{-2}$)↓ | | |
| | | $t+1$ | $t+3$ | $t+5$ | $t+1$ | $t+3$ | $t+5$ | $t+1$ | $t+3$ | $t+5$ | $t+1$ | $t+3$ | $t+5$ |
| Vid2vid (NeurIPS 2018)* [46] | RGB+S | 88.16 | 80.55 | 75.13 | 10.58 | 15.92 | 20.14 | N/A | N/A | N/A | N/A | N/A | N/A |
| Seg2vid (CVPR 2019)* [47] | RGB+S | 88.32 | N/A | 61.63 | 9.69 | N/A | 25.99 | N/A | N/A | N/A | N/A | N/A | N/A |
| FVS (CVPR 2020)* [48] | RGB+S+I | 89.1 | 81.13 | 75.68 | 8.5 | 12.98 | 16.5 | 79.28 | 67.65 | 60.77 | 18.48 | 24.61 | 30.49 |
| SADM (CVPR 2021)* [49] | RGB+S+F | **95.99** | N/A | 83.51 | 7.67 | N/A | 14.93 | 83.06 | 72.44 | 64.72 | 14.41 | 24.58 | 31.16 |
| PredNet (ICLR 2017)* [50] | RGB | 84.03 | 79.25 | 75.21 | 25.99 | 29.99 | 36.03 | 56.26 | 51.47 | 47.56 | 55.35 | 58.66 | 62.95 |
| MCNET (ICLR 2017)* [11] | RGB | 89.69 | 78.07 | 70.58 | 18.88 | 31.34 | 37.34 | 75.35 | 63.52 | 55.48 | 24.05 | 31.71 | 37.39 |
| DVF (ICCV 2017)* [51] | RGB | 83.85 | 76.23 | 71.11 | 17.37 | 24.05 | 28.79 | 53.93 | 46.99 | 42.62 | 32.47 | 37.43 | 41.59 |
| CorrWise (CVPR 2022)* [52] | RGB | 92.8 | N/A | **83.9** | 8.5 | N/A | 15 | 82 | N/A | 66.7 | 17.2 | N/A | 25.9 |
| OPT (CVPR 2022)* [53] | RGB | 94.54 | 86.89 | 80.4 | 6.46 | 12.5 | 17.83 | 82.71 | 69.5 | 61.09 | 12.34 | 20.29 | 26.35 |
| DMVFN (CVPR 2023)* [13] | RGB | 95.73 | **89.24** | 83.45 | 5.58 | 10.47 | 14.82 | **88.53** | **78.01** | **70.52** | 10.74 | 19.27 | 26.05 |
| Ours | RGB | 94.85 | 87.82 | 82.11 | **4.13** | **8.12** | **11.70** | 87.70 | 77.15 | 69.72 | **9.50** | **16.94** | **22.90** |

In Table 5, we compare the proposed method with existing methods on the Cityscapes and KITTI datasets. Following previous research, we report the Multi-scale Structural Similarity Index Measure (MS_SSIM) and LPIPS of the first future frame ($t+1$), the average numbers of the next three future frames ($t+3$) and the average numbers of the next five future frames ($t+5$).

The UCF Sports dataset features sports scenes and thus contains a large amount of fast movements and motion blurs. The leading performance in both Table 3 and 4 validates that the proposed motion graph helps the network better interpret the motion in the input video sequence and facilitate more accurate prediction. Meanwhile, the qualitative result in Figure 5 demonstrates the method's ability to capture and restore intricate image details during the prediction.

For KITTI and Cityscapes datasets, videos are captured from outdoor, large-scale traffic scenarios through cameras with perspective projection, which results in drastic object distortions on both sides of images as well as unstable lighting conditions. Our method matches SOTA performance in terms of quantitative evaluation (see Table 5). In Figure 6, we conducted qualitative comparison on these datasets with two SOTA methods, OPT [53] and DMVFN [13], which are all optical-flow-based.

We noticed that as a non-generative model, our proposed video prediction system may face challenges with occlusions that require the generation of unseen objects. However, it excels in scenarios with scenarios involving occlusions of known objects, as showcased in white walls of the third column in Figure 6. Notably, in scenes with partial obstructions—such as the white wall behind the cyclist, our system adeptly employs multiple motion vectors per pixel to reconstruct occluded areas. This feature

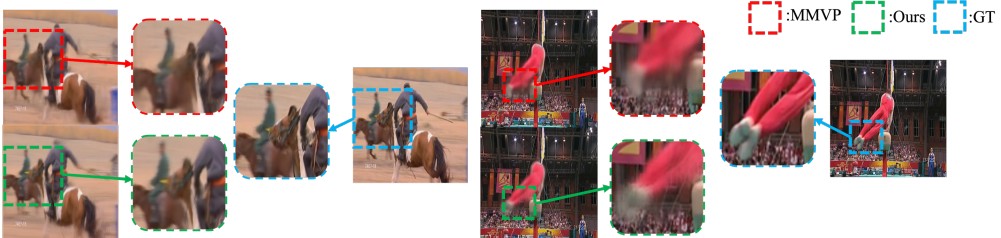

Figure 5: On the UCF Sports dataset, our method recovers richer image details than MMVP [2].

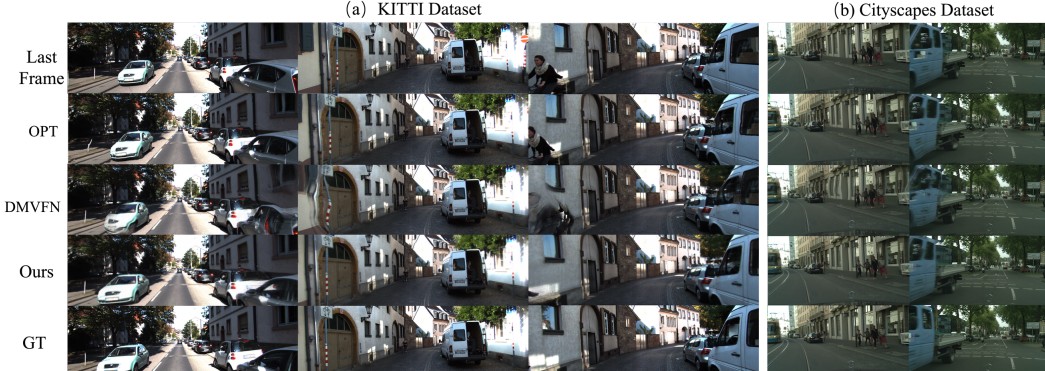

Figure 6: Qualitative comparisons with OPT [53] and DMVFN [13]. Our method maintains the object structures better than both methods while holding a higher motion prediction accuracy.

also supports precise management of object expansion due to perspective projection, as exemplified by the green car in Figure 6's first column.

Moreover, our approach is adept at managing objects exiting the camera's view. By explicitly modeling the motion of image patches, the motion graph predicts when features are about to leave the scene. Thus, any image patches projected to move out of view are not included in the final prediction. This is demonstrated in the 1st, 3rd, and 4th columns of Figure 6, where objects like a cyclist and the front of a blue truck are shown as moving out of frame. Our method successfully and uniquely captures and represents these movements, unlike how the other methods evaluated.

To assess the compactness of the motion graph, we conduct a model consumption analysis. We compare the proposed method with SOTA methods on each dataset in terms of the model size and maximum running GPU memory (the system configurations are identical to the models that generate the numbers in the corresponding tables above). Examination of Table 6 reveals that our method shows robust advantages than SOTA methods in minimizing model consumption and saving resources.

Table 6: Model consumption analysis on three datasets, compared with the SOTA methods.

| Dataset | Image Resolution | Input Frame | Method | Model Size | GPU Memory |
|---------|------------------|-------------|--------|------------|------------|
| UCF Sports | $512 \times 512$ | 4 | MMVP [2] | 2.79M | 4.53GB |
| | | | Ours | **0.60**M | **2.38**GB |
| KITTI | $256 \times 832$ | 2 | DMVFN [13] | 3.56M | 3.79GB |
| | | | Ours | **0.97**M | **0.97**GB |
| Cityscapes | $512 \times 1024$ | 2 | DMVFN [13] | 3.56M | 7.41GB |
| | | | Ours | **0.97**M | **1.25**GB |

## 4.2 Ablation Studies

We conduct ablation studies on three aspects of the UCF Sports MMVP split: i) the choice of $k$, which defines three attributes of the system, i.e., the number of dynamic vectors each node initially embeds,

the number of temporal forward/backward edges for each node, and the number of dynamic vectors to be decoded from the upsampled motion features; ii) the number of views used for multi-view motion graph construction; and iii) the composition of the node's initial features.

Table 7: Ablation study on the value of $k$.

| $k$ | SSIM ↑ | PSNR ↑ | LPIPS ↓ | Memory ↓ |
|---|---|---|---|---|
| 1 | 0.9199 | 29.84 | 0.0868 | 0.98G |
| 5 | 0.9273 | 30.19 | 0.0966 | 1.61G |
| 8 | 0.9285 | 30.25 | 0.0934 | 2.07G |
| 10 | 0.9314 | 30.49 | 0.0823 | 2.38G |
| 20 | 0.9319 | 30.58 | 0.0809 | 3.96G |

Table 8: Ablation study on the number of motion graph views and on the location feature $\mathbf{f}_{loc}$.

| View Number | $\mathbf{f}_{loc}$ | SSIM ↑ | PSNR ↑ | LPIPS ↓ |
|---|---|---|---|---|
| 1 | ✓ | 0.9058 | 28.53 | 0.1205 |
| 2 | ✓ | 0.9158 | 29.47 | 0.1064 |
| 4 | ✓ | 0.9314 | 30.52 | 0.0832 |
| 4 | ✗ | 0.9247 | 30.27 | 0.1008 |

**The choice of** $k$: Adjustments to the value of $k$ and subsequent evaluations reveal a consistent, monotonic increase in both performance and GPU memory usage, as documented in Table 7. Notably, the gain rate diminishes as $k$ increases. This observation aligns with the intuition that increasing the value of $k$ will bring more temporal correspondences to the system. However, once the motion information nears saturation, the incremental benefit of further increasing $k$ becomes less significant. More ablation studies related to $k$ can be found in Section A.4.

**The number of graph views**: As Section 3.2 mentioned, using multi-view graphs can enhance motion prediction accuracy. To validate this claim, we reduce the number of views used in the system and test the system on the UCF Sports dataset. The first three rows in Table 8 show that increasing the number of graph views indeed improves the system performance.

**Location feature** $\mathbf{f}_{loc}$: We incorporate the tendency feature $\mathbf{f}_{tdc}$ (directly related to motion) and a location feature $\mathbf{f}_{loc}$ to initialize the node features. This inclusion hypothesizes that an image pixel's motion patterns may also be influenced by its spatial position. Empirical evidence supporting this hypothesis is documented in the last two rows of Table 8. There is a notable decline in performance when the location feature $\mathbf{f}_{tdc}$ is excluded from the node feature initialization. In Section A.7, we further visualize the location features and observe that such feature may reflect the camera projection pattern, providing substantial information to the following motion reasoning task.

## 5 Conclusion & Limitation

This work presents the motion graph for video prediction and a novel pipeline built upon it. We focus on balancing *representative ability* and *compactness* to optimize efficiency. The motion graph encapsulates complex motion information hidden in video sequences into a more manageable format, and achieves encouraging results: it matched or outperformed SOTA on three well-known datasets with a considerably smaller model size and less GPU running memory. Beyond the results, the motion graph and its associated video prediction pipeline set a foundation for further research and optimization in the field of video prediction, thus suggesting a promising direction for researchers seeking to balance performance and resource efficiency in the evolving domain of video processing.

**Limitation.** There is no significantly improvement in terms of the inference speed. The current version of our model is slower than DMVFN, which is specifically optimized for this aspect. Future efforts will focus on accelerating inference while maintaining the model's compactness and efficiency. Additionally, video prediction remains particularly challenging in scenarios involving sudden or unpredictable motions, which our system occasionally fails to capture, as highlighted in Appendix A.6. These instances, where the model struggles with abrupt actions not readily discernible from the input frames, underscore the need for further enhanced video understanding abilities. Addressing these challenges presents a valuable direction for future research.

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

# A    Appendix

## A.1    Implementation Details

We implement the video prediction system using PyTorch [54] and conduct end-to-end training on a single NVIDIA A100 GPU. We use AdamW optimizer [55] during the training. The initial learning rate is set to $1e^{-3}$ and decayed to $1e^{-5}$ following a cosine decay scheduler [56]. There are only a few hyperparameters to adjust for the system when training on different datasets. The adjustments are mainly based on the resolution of the frame. Those hyper-parameters include i) image feature length (Image feat.), which is the parameter for the image encoder; ii) Tendency feature length (Tendency feat.), which is the length of the tendency feature in node initialization; iii) location feature length (Location feat.), which is fixed to 4 for all datasets, indicating the length of the location feature in node initialization; iv) the number of graph views, indicating view number in motion graph feature learning; v) $k$, indicates the number of the dynamic vectors embedded in each node, the number of the temporal edges per node and the output dynamic vectors per pixel; vi) training epoch is the training related parameters; vii) the reconstruction loss, which follows the popular setting of the SOTA methods on each dataset. In Table 9, we demonstrate the hyper-parameter setting for each dataset.

Please note that we did not especially tune the parameters for each dataset. When adjusting the parameters, we consider more about the training efficiency instead of the performance. Therefore, our setting is likely *not* the optimal choice. For example, in DMVFN [13], the training on Cityscapes and KITTI are both 300 epochs, we observe that our system can achieve comparable performance with only 100 and 200 epochs respectively, we thus stay with this configuration.

| Dataset | Image feat. | Tendency feat. | Location feat. | Number of Graph views | $k$ | Epoch | Loss |
|---|---|---|---|---|---|---|---|
| UCF Sports | 16 | 16 | 4 | 4 | 10 | 300 | MSE |
| Cityscapes | 16 | 32 | 4 | 4 | 10 | 100 | L1 + Lpips |
| KITTI | 16 | 32 | 4 | 4 | 8 | 200 | L1 + Lpips |

Table 9: Hyper-parameter setting for each dataset.

## A.2    Network Architecture

The proposed video prediction system includes three major components, the image encoder, the motion graph interaction module, and the motion upsampler. Here we demonstrate the detailed architecture of each component for reproduction needs.

**Image Encoder**: Figure 7 shows the inner structure of the image encoder in the proposed system. $C_{img}$ is related to the image feature length in Table 9. Each convolution layer will come with a Leaky ReLU layer [57] as the activation layer.

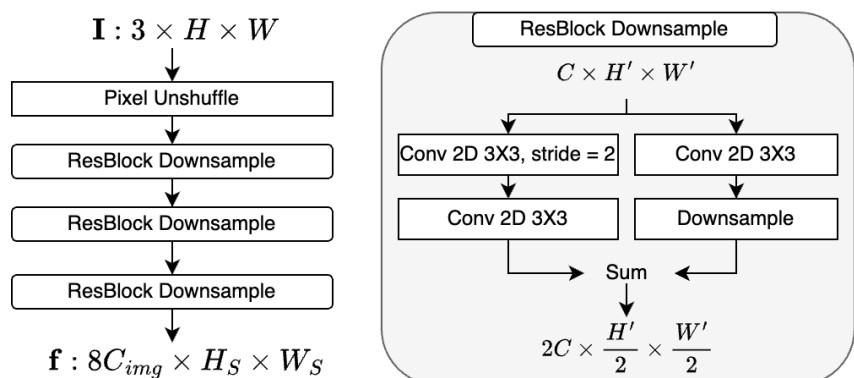

Figure 7: Architecture of the image encoder

**Motion Graph Interaction Module** In Figure 8 we demonstrate the inner structure of the spatial and temporal message passing in the motion graph interaction module. $C_{node}$ equals the sum of the tendency feature length and the location feature length in Table 9.

**Spatial:**

$$V : N \times C_{node} \xrightarrow[Reshape]{} V : C_{node} \times H_S \times W_S \rightarrow \boxed{\text{Conv2D 3X3}} \xrightarrow[Reshape]{} V' : N \times C_{node}$$

**Temporal:**

$$V_{pred} : N_{pred} \times C_{node} \longrightarrow \boxed{\text{Linear Projection}} \rightarrow V_{pred} : N_{pred} \times C_{node}$$
$$\downarrow \times W_{pred} : N_{pred} \times 1$$
$$V_{suc} : 1 \times C_{node} \longrightarrow \oplus \rightarrow \boxed{\text{Linear Layer}} \rightarrow V'_{suc} : 1 \times C_{node}$$

Figure 8: Inner structure of spatial and temporal block in motion graph interaction module

**Motion Upsampler** Figure 9 illustrates the inner structure of the motion upsampler as well as the motion decoder. The implementation of the decoder is a single 2D convolution layer with a kernel size of 1.

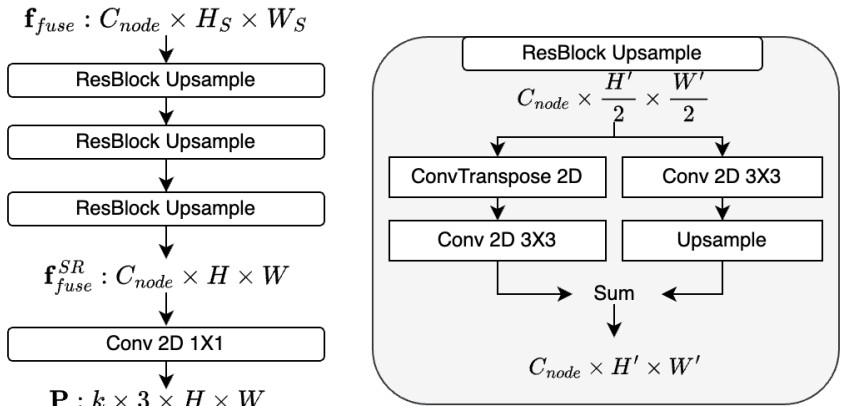

Figure 9: Inner structure of the motion upsampler and the motion decoder.

## A.3   Additional Quantitative Evaluation

For comparison convenience of the future works, we list the extra metrics of cityscapes and KITTI datasets in Table10, i.e. ssim and psnr.

Table 10: Evaluation on Cityscapes [39] and KITTI [37] datasets. "RGB", "F", "S" and "I" denote video frames, optical flow, semantic map, and instance map; $t + 3$ and $t + 5$ respectively indicate the average performance of the next 3 and 5 frames. Results with * are copied from [13].

| Method | Input | Cityscapes | | | | | | KITTI | | | | | |
| | | SSIM($\times 10^{-2}$)↑ | | | PSNR↑ | | | SSIM($\times 10^{-2}$)↑ | | | PSNR↑ | | |
| | | $t+1$ | $t+3$ | $t+5$ | $t+1$ | $t+3$ | $t+5$ | $t+1$ | $t+3$ | $t+5$ | $t+1$ | $t+3$ | $t+5$ |
|---|---|---|---|---|---|---|---|---|---|---|---|---|---|
| DMVFN (CVPR 2023)* [13] | RGB | 87.45 | 78.75 | 63.95 | 28.81 | 25.47 | 23.61 | 72.69 | 62.52 | 57.05 | 22.74 | 20.16 | 18.70 |
| Ours | RGB | 92.30 | 86.51 | 83.16 | 28.43 | 25.39 | 23.80 | 77.91 | 69.55 | 65.05 | 21.71 | 19.44 | 18.18 |

## A.4   Extensive Ablation Study

In this section, we add two ablation studies to help the audience better interpret the design of the motion graph.

**Number of the predicted dynamic vectors per pixel:** In the proposed system, we set the number of the predicted dynamic vectors per pixel to $k$, which is identical to the number of the dynamic vectors embedded by each node and the temporal edge of each node. This design ensures the flexibility of the predicted motion to have multiple modes compared to the optical-flow-based method which

only allows each pixel to have a single future motion. The comparison between the first two rows of Table 11 showcase that allowing multiple predicted dynamic vectors can largely improve the performance. Meanwhile, if we control the number of the predicted dynamic vectors, as demonstrate by the comparison between the first and third row of the Table 11, we see that when the motion graph embeds more past motion modes, the performance will also has significant improvements.

| $k$ | Predicted Vectors # | Full set | | | Hard (SSIM $< 0.6$) | | | Memory $\downarrow$ |
|---|---|---|---|---|---|---|---|---|
| | | SSIM $\uparrow$ | PSNR $\uparrow$ | LPIPS $\downarrow$ | SSIM $\uparrow$ | PSNR $\uparrow$ | LPIPS $\downarrow$ | |
| 1 | 1 | 0.8742 | 26.34 | 0.1527 | 0.5394 | 17.34 | 0.5032 | 0.98G |
| 1 | 10 | 0.9199 | 29.87 | 0.1042 | 0.6408 | 19.44 | 0.3706 | 1.97G |
| 10 | 1 | 0.9212 | 30.00 | 0.0877 | 0.6546 | 19.63 | 0.3261 | 1.38G |

Table 11: Ablation study on the number of the predicted vectors. The experiments are conducted on UCF Sports MMVP splits. The listed results are from the models trained for 100 epochs (models were all trained for 300 epochs in the main manuscript).

**Motion Graph Interaction Module** The design of the motion graph interaction module are following the intuition that both spatial connection and temporal connection should benefit the graph learning. Here we also show the experimental results in Table 12 that both spatial and backward edges are beneficial to the final performance.

| Spatial | Backward | PSNR $\uparrow$ | MS-SSIM $\times 100 \uparrow$ | LPIPS $\times 100 \downarrow$ |
|---|---|---|---|---|
| $\times$ | $\times$ | 21.55 | 87.06 | 9.85 |
| $\checkmark$ | $\times$ | 21.64 | 87.25 | 9.83 |
| $\checkmark$ | $\checkmark$ | 21.71 | 87.70 | 9.50 |

Table 12: Ablation study on graph interaction module. The experiments are conducted on KITTI and metrics show evaluation on the $t + 1$ results.

## A.5 Extensive Discussion on Experiment setting

Our research on video prediction has identified key differences between systems designed for short-term and long-term predictions. Short-term systems typically use fewer frames to predict the immediate next few frames, while long-term systems are tasked with forecasting an extensive sequence of future frames. The design logic and objectives of these systems, thus, vary significantly. Table 13 demonstrates the differences with more details. It is easy to notice that the recent works of

| | Short-term Video Prediction | Long-term Video Prediction |
|---|---|---|
| Prediction Length | Limited, usually one frame | Much longer, $\gg 10$ frames |
| Video Resolution | Up to 4K | Usually smaller, up to $256 \times 256$ |
| Common Dataset | UCF 101, UCF Sports, KITTI, Cityscapes, SJTU4k, Vimeo, DAVIS | KTH, moving MNIST, BAIR, cropped Cityscapes |
| Recent works | SIMVP[1], MMVP[2], STRPM[15], STIP[16], DMVFN[13] | MVCD[58], MaskViT[8],VIDM[59],ExtDM[60] |
| Objectives | High resolution videos, pixel-level accuracy, real-time application | Conditional video generation, semantic-level accuracy |

Table 13: Experiment setting comparison between short-term and long-term video prediction task

long-term video prediction usually involve with diffusion-based generative models, which is designed to feed the need of predicting long videos. The evaluation of such methods usually emphasize on if the predicted video is semantically correct given the input video frame(s). While in this study, we emphasize our method's advanced motion modeling and significant reduction in computational costs, essential for short-term prediction of high-resolution videos. Our system is tailored for short-term video prediction, with evaluations conducted along this line of work.

We plan to further exploit the motion graph's potential as an efficient motion representation tool and develop advanced, motion graph-based systems for long-term video generation in our next work.

## A.6 Failure case demonstration

The video prediction is always a challenging problem. Especially for those video sequences with abrupt motion which can be hardly indicated by the previous video frames. The proposed method

Frame 1 to 4                          Frame 5 (GT)    Prediction

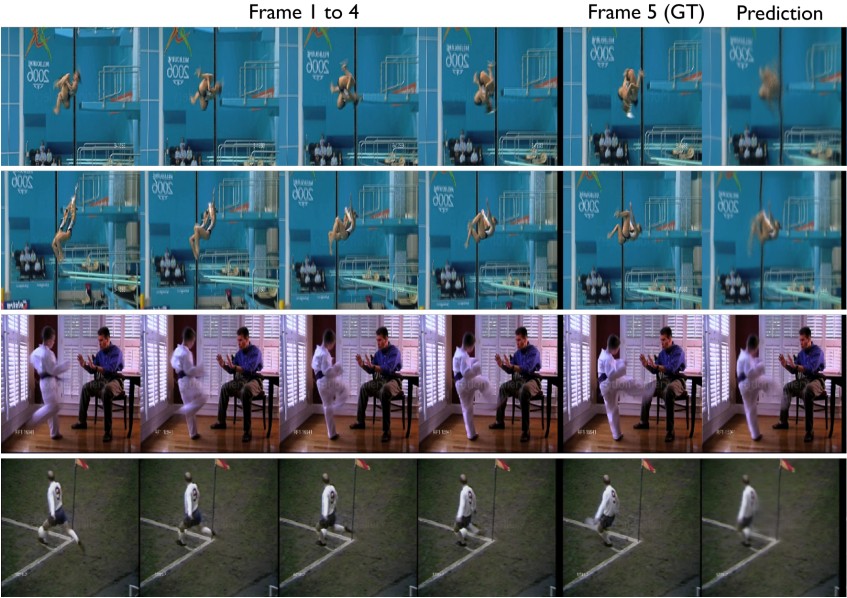

Figure 10: Failure cases in UCF Sports Dataset

formulates the video prediction as a motion prediction problem and outperforms most of the existing methods by using motion graph to better capture the motion hints from the input frames. However, when evaluating the qualitative results, we still find some failure cases that require additional research efforts to solve. In Figure 10, we showed typical failure cases in UCF Sports dataset. We notice that most of the failures cases are in the action of kicking and diving, which usually include fast, unpredictable motion that requires stronger video understanding capability of the model.

### A.7  Node Feature Visualization

To better understand the initialization of the node embedding, here we visualize the tendency feature and the location feature. We first extract the tendency feature and location feature of each node in the motion graph and apply a K-means clustering to the extracted features. For the tendency feature, we set the cluster number to 2; and for the location feature, we set the cluster number to 4 for better visualization.

From Figure 11, we see that using the learned tendency feature, the system should be able to distinguish the dynamic areas from the static areas. If we further enlarge the cluster number, we can see more clearly that the tendency features embed the different motion patterns of each feature patch in the frame. For the location feature, in the paper, we have shown that removing the location feature from the node initialization will result in a performance drop. From Figure 12 we observe that the location feature may contain information that is related to the camera projection mode. For cityscapes and the KITTI, which use wide-range cameras, the clustering pattern of the location feature is very different from the UCF Sports whose projection mode is possible to be orthogonal projection.

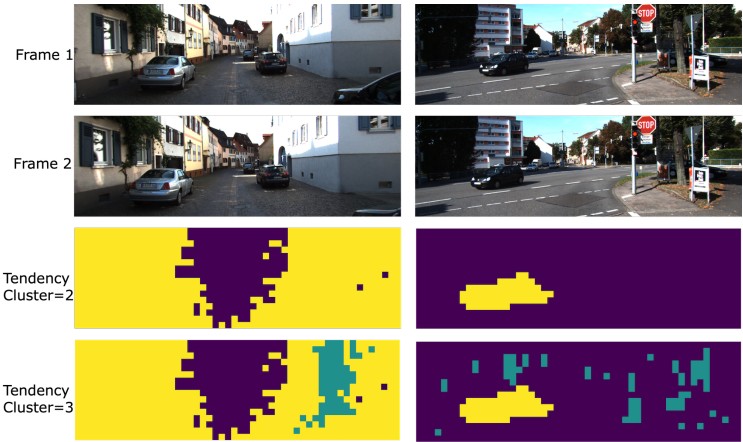

Figure 11: Tendency feature visualization using KITTI dataset

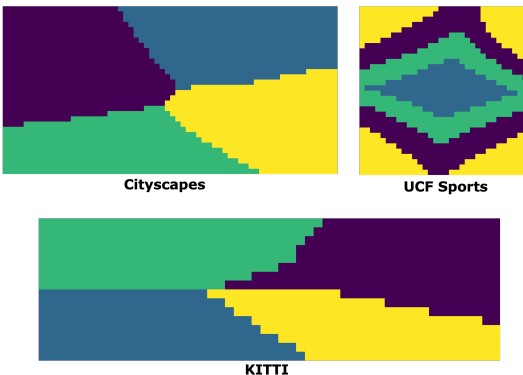

Figure 12: Locaiton feature visualization on three datasets.

