# OpenReview forum: "Motion Graph Unleashed: A Novel Approach to Video Prediction"
_NeurIPS.cc/2024/Conference — NeurIPS 2024 poster_

### Official Review · Reviewer_V8rg · 2024-07-02

**Soundness:** 3
**Presentation:** 4
**Contribution:** 2
**Rating:** 5
**Confidence:** 4

**Summary:**

The paper proposes the motion graph method in predicting the video frames by exploring the spatio-temporal relations among frames from the limited data.

**Strengths:**

It is significant to propose methods of few-shot prediction techniques in video inputs.

**Weaknesses:**

Please provide more details about the setup of the datasets such as KITTI and Cityscapes that are not originally used for testing video prediction.

The results on Cityscapes and KITTI datasets based on LPIPS, SSIM, PSNR are needed to be explained in detail in the evaluation methods either in the main text or Appendix.

**Questions:**

.

**Limitations:**

The authors address the limitations briefly.

---

> ### Author Rebuttal · Authors · 2024-08-06
>
> Question 1:  More details about the setup of the datasets such as KITTI and Cityscapes that are not originally used for testing video prediction.
>
> Answer 1: We follow the experiment setup from previous works which originated from *Wu, Yue, et al. "Future video synthesis with object motion prediction". Proceedings of the IEEE/CVF Conference on Computer Vision and Pattern Recognition. 2020.* We use the data preprocessing code from this repo, https://github.com/hzwer/CVPR2023-DMVFN/blob/main/utils/, to generate the train/val data.
>
> ---
>
> Question 2: The results on Cityscapes and KITTI datasets based on LPIPS, SSIM, PSNR are needed to be explained in detail in the evaluation methods either in the main text or Appendix.
>
> Answer 2: Thank you for the suggestion. We will add the calculation details in the appendix.

---

> > ### Comment · Reviewer_V8rg · 2024-08-08
> >
> > I have read the rebuttal of the authors.
> >
> > I have seen the authors' main point on generating their literature from the previous work using KITTI and Cityscapes datasets for video prediction. I appreciate the effort put into this and hope that further researchers can benefit from that literature in the future and the related works multiply based on video prediction.
> >
> > As I tend to accept this work now, I hope to see the evaluation methods of LPIPS, SSIM, and PSNR on Cityscapes and KITTI datasets in the appendix in your revised version of the manuscript.

---

> > > ### Author Response · Authors · 2024-08-08
> > >
> > > Thank you for the valuable suggestion. We recognize that few existing studies report PSNR and SSIM metrics for evaluations on the KITTI and Cityscapes datasets. To support future research in the video prediction community, we will add a new table and discussion to Appendix A.3, detailing these metrics for our method. Additionally, we will compile and compare available results from other methods. We also plan to release the result images from these two datasets to enable more convenient quantitative and qualitative comparisons for future researchers.

---

### Official Review · Reviewer_Vf8f · 2024-07-09

**Soundness:** 3
**Presentation:** 4
**Contribution:** 3
**Rating:** 7
**Confidence:** 3

**Summary:**

The paper introduces a graph-based methods to predict video frames through motion prediction. The proposed motion graph captures complex spatial-temporal relationships by transforming video frame patches into interconnected graph nodes. This method improves performance and reduces computational costs compared to SOTA methods on UCF Sports, KITTI and Cityscapes.

**Strengths:**

1. The paper introduces a novel motion graph representation, combining graph theory with video prediction to overcome existing limitations.
2. It has comprehensive evaluations across multiple datasets and achieve superior performance than prior methods.
3. The manuscript is well organized and has detailed experimental results.

**Weaknesses:**

1. Slower inference speed compared to optimized methods like DMVFN.
2. Struggles with predicting abrupt or unpredictable motions.
3. Unclear if the motion graph captures hierarchical / semantic feature of motions.
4. Does not address the stochastic nature of human videos.
5. Limited diversity in evaluation datasets, focusing mainly on sports and driving scenarios.

**Questions:**

1. Do the authors see a future where such work might be useful for improving video generative models?
2. Can the authors explain what the weight components are for? It seems that the dynamic vectors can represent motion even without the weighted component. If it does represent something meaningful, can the authors show what they represent after training? For example, might it correspond to the change in depth?
3. How does the motion graph approach compare to recent transformer-based video prediction methods in terms of performance and efficiency?
4. How sensitive is the method to the choices made in graph construction (e.g., number of neighbors, edge types)?
5. How does the motion graph approach scale with longer video sequences? Is there a practical limit to the number of frames it can handle efficiently?

**Limitations:**

The authors briefly mention limitations regarding inference speed and handling sudden motions. However, they could provide a more comprehensive discussion of potential limitations, such as scalability to longer videos or generalization to diverse video types.

---

> ### Author Rebuttal · Authors · 2024-08-06
>
> Question 1: Do the authors see a future where such work might be useful for improving video generative models?
>
> Answer 1: Yes. This work uses video prediction task as an example to validate the efficiency and effectiveness of motion graph as a comprehensive video motion representation. In the future, we will explore motion graph's potential as an efficient motion representation tool, developing advanced motion graph-based systems for long-term video generation.
>
> ---
>
> Question 2: Can the authors explain what the weight components are for? It seems that the dynamic vectors can represent motion even without the weighted component. If it does represent something meaningful, can the authors show what they represent after training? For example, might it correspond to the change in depth?
>
> Answer 2:  After training, the graph construction process uses weights from the cosine similarity scores between pairs of image patches to determine their connectivity. Similarly, the weights output by the motion upsampler function as confidence scores for each predicted motion vector. This weighting strategy offers several advantages:
>
> a) Graph Construction Robustness: In the motion graph, each image patch ideally connects to $k$ patches in the subsequent frame. If a patch is occluded in the next frame, selecting the top $k$ connections could lead to random, inaccurate connections. The weighting strategy helps identify and minimize these erroneous connections.
>
> b) Selective Information Aggregation: During graph learning, this approach allows the model to focus on high-confidence correspondences and reduce the influence of ambiguous connections, enhancing the overall learning effectiveness.
>
> c) Enhanced Frame Synthesis: During the synthesis of future frames, multiple source pixels may correspond to a single pixel in the future frame. The weighted approach allows for strategic aggregation of these source pixels, improving the quality of the resulting image.
>
> ---
>
> Question 3: How does the motion graph approach compare to recent transformer-based video prediction methods in terms of efficiency and performance?
>
> Answer 3: We appreciate the reviewer initiating this discussion. Both the motion graph approach and transformer-based video prediction methods (e.g. MaskViT) share the fundamental goal of predicting unknown relational information by modeling existing temporal correspondences between adjacent frames. However, significant differences in efficiency, performance, and adaptability set them apart:
>
> a) Efficiency: Transformer-based methods create a complete graph for all feature patches between two adjacent frames, employing complex models that require a large number of parameters. This often leads to high computational demands, making these methods less suitable for high-resolution video processing. In contrast, the motion graph approach constructs a sparser graph, with edge weights determined by feature similarities from a CNN-based image encoder. This results in a more lightweight model. Furthermore, motion graphs leverage Graph Convolutional Networks (GCNs) composed of stacked linear layers, enabling efficient learning from the graph structure.
>
> b) Performance: The efficiency differences make direct performance comparisons with transformer-based methods challenging. Currently, few transformer-based approaches are tested on the high-resolution datasets discussed in this manuscript, leaving performance comparisons inconclusive.
>
> c) Adaptability to Different Video Resolutions: A major drawback of transformer-based methods is their lack of flexibility with varying input resolutions during testing. They struggle to process videos of different resolutions effectively. Conversely, our motion graph approach employs a CNN-based image encoder and motion upsampler, offering greater adaptability to various video resolutions. This adaptability enhances the suitability of the motion graph method for real-world applications.
>
> ---
>
> Question 4: How sensitive is the method to the choices made in graph construction (e.g., number of neighbors, edge types)?
>
> Answer 4: Our ablation study (detailed in Table 7) reveals a monotonic performance increase as the value of $k$ increases, although the rate of gain diminishes with larger $k$ values. This suggests that performance is more sensitive to changes in $k$ when $k$ is small, as it more closely resembles an optical flow-based method. However, as $k$ increases and motion information approaches saturation, the impact of $k$ lessens. Additionally, Table 12 in Appendix A.4 demonstrates that including both spatial and backward edges enhances model performance.
>
> ---
>
> Question 5: How does the motion graph approach scale with longer video sequences? Is there a practical limit to the number of frames it can handle efficiently?
>
> Answer 5:  Longer video sequence can add more nodes and edges to the motion graph, which will result in an increase in the parameter number and computational overhead of motion graph learning. However, in terms of graph construction and feature extraction, the impact of video sequences length will be less, since we extract each image frame individually through an image encoder. Lastly, the practical limit to the number of frames a motion graph approach can handle efficiently is typically dependent on the hardware constraint. The available GPU memory significantly influences how many frames can be effectively managed.

---

### Official Review · Reviewer_Mw8x · 2024-07-13

**Soundness:** 4
**Presentation:** 2
**Contribution:** 4
**Rating:** 7
**Confidence:** 4

**Summary:**

This paper proposes a motion-based method for video prediction. They design a new motion representation named motion graph that transforms patches of video frames into interconnected graph nodes. The proposed video prediction pipeline, empowered by the motion graph, exhibits substantial performance improvements and cost reductions. Experiments on UCF Sports, KITTI, and Cityscapes are conducted.

**Strengths:**

+ Motion-based video prediction is quite an interesting topic and benefits the research community with an affordable training cost.
+ The evaluation of both efficiency and effectiveness is conducted.
+ The paper is well-structured.

**Weaknesses:**

- Motivation and Occlusion / Out-of-View cases. As the authors claimed in lines 33-37 and Figure 1, existing methods struggle to effectively handle occlusion cases. However, the paper does not provide a corresponding evaluation to demonstrate the effectiveness of the proposed motion graph in complex situations involving occlusion or out-of-view scenarios. This is a significant omission, as these challenges are prevalent in real-world video prediction tasks. Furthermore, based on my understanding of the work, the proposed motion graph may not adequately address these problems. Specifically, occluded and out-of-view object/background do not appear to be considered in the graph, as the connections and features captured in the motion graph are derived from the visible parts of the targets. This raises concerns about the method's ability to maintain accurate predictions when dealing with occlusions or objects moving out of the camera's field of view. The authors should provide a thorough discussion that includes both the theoretical principles and empirical evidence.

- Video prediction evaluation setting. While the proposed motion graph method shows promising results in predicting a few frames (up to 10), the paper lacks an evaluation of its performance in predicting a larger number of frames. Other video prediction works [A, B, C] have evaluated their methods on longer prediction horizons, providing a more comprehensive assessment of their models' robustness and accuracy.

[A] ExtDM: Distribution Extrapolation Diffusion Model for Video Prediction, CVPR24

[B] Efficient Video Prediction via Sparsely Conditioned Flow Matching, ICCV23

[C] MCVD: Masked Conditional Video Diffusion for Prediction, Generation, and Interpolation, NeurIPS22

- Comparison with MMVP. The proposed motion graph method appears to be quite similar to the motion matrix used in MMVP, which also provides a dense connection among frames. Although I appreciate the discussion provided in the related work, the authors do not sufficiently clarify the differences and advantages of the proposed method compared to MMVP. This lack of distinction makes it difficult to understand the unique contributions and potential improvements offered by the motion graph.

- Comparison with other motion-cues-guided video prediction methods. The paper lacks a comparison with other motion-cues-guided video prediction methods, such as MotionRNN, which also leverages a vector-based motion representation. The authors should explain how the feature warping mechanism in MotionRNN compares to the feature processing in the motion graph. What are the potential benefits or drawbacks of each approach? Are there specific scenarios where the motion graph's approach to feature handling provides a clear advantage?

- Lack of a component-wise ablation. The reviewer noticed that there is no component-wise ablation study for motion feature learning, upsampling, decoding, image wrapping, and interaction modules. It makes it difficult for the audience to identify which parts of the methodology are effective and contribute most to the overall performance. The authors should conduct a thorough ablation study that isolates and evaluates the impact of each component of their proposed method.

- Complexity of the proposed method and presentation. Actually, the proposed method is somehow non-trivial, making this work hard to follow. Accurate presentation can make the work easier to understand.

**Questions:**

See weakness

**Limitations:**

The authors indicate limitations in the conclusion section.

---

> ### Author Rebuttal · Authors · 2024-08-06
>
> Q1. Motivation and occlusion / out-of-view cases.
>
> A1: As a non-generative model, our proposed video prediction system may face challenges with occlusions that require the generation of unseen objects. However, it excels in scenarios with scenarios involving occlusions of known objects, as showcased in the third column of Figure 6, where our method outperforms existing SOTA methods. Notably, in scenes with partial obstructions—such as the white wall behind the cyclist—our system adeptly employs multiple motion vectors per pixel to reconstruct occluded areas. This feature also supports precise management of object expansion due to perspective projection, as exemplified by the green car in Figure 6's first column.
>
> Moreover, our approach is adept at managing objects exiting the camera’s view. By explicitly modeling the motion of image patches, the motion graph predicts when features are about to leave the scene. Thus, any image patches projected to move out of view are not included in the final prediction. This is demonstrated in the 1st, 3rd, and 4th columns of Fig 6, where objects like a cyclist and the front of a blue truck are shown as moving out of frame. Our method successfully and uniquely captures and represents these movements, unlike how the other methods evaluated.
>
> ---
>
> Q2. Video prediction evaluation setting.
>
> A2: Thank you for initiating this discussion. Our research on video prediction has identified key differences between systems designed for short-term and long-term predictions. Short-term systems typically use fewer frames to predict the immediate next few frames, while long-term systems are tasked with forecasting an extensive sequence of future frames. The design logic and objectives of these systems, thus, vary significantly. For a more detailed comparison, we have outlined these distinctions in the table below:
> |Task type|Output length|Resolution|Dataset|Recent work|Objectives|
> |---|---|---|---|---|---|
> |Short-term|Short|Up to 4K|UCF 101, UCF Sports, KITTI,  Cityscapes, SJTU4k, Vimeo, DAVIS|SIMVP, MMVP, STRPM,  STIP, DMVFN|High resolution videos, pixel-level accuracy, real-time application|
> |Long-term|Long ($\gg 10$)|Up to $256\times 256$|KTH, moving MNIST,  BAIR, cropped Cityscapes|MVCD, MaskViT,VIDM,ExtDM|Conditional video generation, semantic-level accuracy|
>
> In this study, we emphasize our method's advanced motion modeling and significant reduction in computational costs, essential for short-term prediction of high-resolution videos. Our system is tailored for short-term video prediction, with evaluations conducted along this line of work. We plan to further exploit the motion graph's potential as an efficient motion representation tool and develop advanced, motion graph-based systems for long-term video generation in our next work.
>
> ---
>
> Q3: Comparison with MMVP.
>
> A3: We highlight in the manuscript three major differences to MMVP.
> |Method|Motion representation|Prediction module architecture|Motion & Appearance Composition method|
> |---|---|---|---|
> |MMVP|Motion Matrix   ($H\times W\times H\times W$)|3D Convolution based network|Downsampled-feature-level matrix multiplication|
> |Motion Graph (Ours)|Motion Graph   ($H\times W\times k\times 3$) |Graph convolution network|Pixel-level image forward warping|
>
> The advantages of our method are evidenced by its performance in various tests:
>
> a) Our sparser motion representation allows the system to process videos with **higher resolutions using fewer computational resources**. For instance, Table 6 indicates that GPU consumption for our method is 52% of that required by MMVP.
>
> b) Our graph-based motion prediction module **aggregates relational information from motion embeddings more efficiently**, resulting in a lighter and more effective model. As shown in Tables 4 and 6, our method requires only 21% of the model size of MMVP while improving the LPIPS metric by 22%.
>
> c) **Pixel-level image warping** maintains critical appearance information from existing frames, significantly boosting detail recovery. Figure 5 demonstrates this by comparing the clarity of details like the horse’s facial features, the rider’s face, and the athlete’s feet, with our method outperforming others in detail retention.
>
> ---
>
> Q4: Comparison with other motion-cues-guided video prediction methods.
>
> A4:  Although MotionRNN also utilizes motion vectors, there are several fundamental differences:
>
> a) MotionRNN employs local $3\times3$ motion filters, which are limited in handling large-scale motion;
>
> b) MotionRNN does not account for the interaction of motion across different spatial and temporal regions, a feature that our motion graph learning process emphasizes;
>
> c) MotionRNN performs feature-level warping and our approach conducts pixel-level warping, providing finer detail.
>
> These distinctions contribute to the lesser robustness of MotionRNN in handling high-resolution videos with complex motion scenarios, as detailed in Tab 3 of the manuscript. We have reviewed various motion-based video prediction methods in Sec 2. Moving forward, we plan to conduct a further literature review to expand and update this section accordingly.
>
> ---
>
> Q5: Lack of component-wise ablation study.
>
> A5: The components of motion upsampling, motion decoding, and image warping modules are each essential to the functionality of the system. So we have opted not to isolate each component for ablation studies. Instead, we focus the ablation analysis on the composition of motion features: please see details in Table 8 of the manuscript. Appendix A.4 also provides additional ablation studies, with Table 12 specifically exploring the effects of various interaction modules.
>
> ---
>
> Q6. Presentation of this complex method.
>
> A6: Thank you for this comment. To more effectively present our proposed method, for it to be more comprehensible and accessible, in the revised manuscript we worked on the clarity of descriptions and incorporated additional visual explanations.

---

> > ### Comment · Reviewer_Mw8x · 2024-08-13
> > **Official Comment by Reviewer Mw8x**
> >
> > Thanks for the rebuttal. Most of my concerns are addressed.
> >
> > I'd like to see the discussions about occlusions, long-term VP, and other related methods in the revised version.

---

> > > ### Author Response · Authors · 2024-08-13
> > >
> > > We deeply appreciate your suggestions and will incorporate a discussion on occlusion, long-term video prediction, and comparisons with other methods into the revised manuscript.

---

### Official Review · Reviewer_gkX2 · 2024-07-13

**Soundness:** 3
**Presentation:** 2
**Contribution:** 3
**Rating:** 5
**Confidence:** 2

**Summary:**

The authors propose a Motion Graph for predicting future video pixels. To achieve this, they introduce three modules: 1. Motion Graph Node Construction; 2. Edge Construction; 3. Graph Interaction Module; 4. Video Prediction Model. The first three modules encode patches and their interactions in a frame, while the fourth module decodes future frame pixels. Experiments on UCF Sport, KITTI, and CityScape demonstrate the method's high efficiency and construction quality.

**Strengths:**

1. The author propose to predict future pixels from motion graph perspective, to achieve this, they propose motion graph constuction and graph decoding modules.

2. The method demonstrate high computation efficieny and good reconstruction quality on few video benchmarks.

**Weaknesses:**

1. It's better to present an overview figure for the framework. It's not easy to grasp the overall structure by reading about the four separate modules.

2. Figures 2-4 have fonts that are too small, making them hard to read.

**Questions:**

1. Which backbone are used to extract features during motion graph construction. Does the method robust to different backbones?
2. Does the prediction quality decrease with respect to resolutions? is it possible to analyze prediction quality with large resoultion settings.

**Limitations:**

The authors discuss and partially address the limitations of the paper.

---

> ### Author Rebuttal · Authors · 2024-08-02
>
> Question 1: Which backbone are used to extract features during motion graph construction. Does the method robust to different backbones?
>
> Answer 1: ResNet was used as the backbone of our image encoder to extract features during motion graph construction. In the manuscript submission, Figure 7 of Appendix A.2 illustrates the network architecture of the image encoder. Our experiments demonstrate the robustness of our design across different image encoder architectures. We selected a comparatively lightweight model to ensure both model compactness and system computational efficiency.
>
> ---
>
> Question 2: Does the prediction quality decrease with respect to resolutions? is it possible to analyze prediction quality with large resolution settings.
>
> Answer 2: To evaluate the model's performance on high-resolution images, we tested our method on the SJTU4K dataset, which has a resolution of $2160\times3840$. Due to the absence of publicly available data splits for direct comparison with existing state-of-the-art (SOTA) methods, we created our own training and validation splits, allocating 80% of the sequences to training and 20% to validation.
>
>
> | Dataset | Resolution | LPIPS $\downarrow$ | MS_SSIM $\uparrow$ |
> |:---:|:---:|:---:|:---:|
> | KITTI | $256\times 832$ | 9.50 | 87.70 |
> | Cityscapes | $512\times 1024$ | 4.13 | 94.85 |
> | SJTU4K | $2160\times 3840$ | 8.22 | 90.70 |
>
>
> While the model's performance on the SJTU4K dataset surpasses its performance on KITTI, which has a much lower resolution, it does not match the performance on Cityscapes, which also has a smaller resolution than SJTU4K. Our findings suggest that factors such as motion complexity, frame rate, and scene complexity have a greater influence on prediction performance than frame resolution alone.

---

### Author Rebuttal · Authors · 2024-08-06

We are grateful for the thoughtful suggestions and comments from the reviewers.

In response, we have implemented several enhancements to the manuscript, including, but not limited to:


a) Enhancing the visual presentation throughout, such as adding an overview figure;


b) Adjusting the text font size of the figures for better readability;

c) Expanding the literature review to include more comprehensive discussions;

d) Clarifying the descriptions of our methods for improved understanding;

e) Providing additional details about the evaluation metrics and the development of the testing datasets.

Please see below for our point-by-point response to the reviewers' feedback. Thank you for your time.

---

### Decision · Program_Chairs · 2024-09-25

**Decision:**

Accept (poster)

**Comment:**

Initially, the reviewers raise some questions regarding the robustness to different backbones, the prediction quality with respect to resolutions, occlusion and out-of-view cases, video prediction evaluation setting, lack of component-wise ablation, sensitivity in graph construction, diversity of metric evaluation, etc. Most of these questions are addressed in the rebuttal and recognized by reviewers. Eventually with the merit of the work, this paper receives two accepts and two borderline accepts, with agreement for acceptance. The AC recommends to accept. Authors are encouraged to revise the paper according to the reviews, including adding a discussion about occlusion, long-term video prediction, and the comparison with other methods, etc.